# Batch Fabrication of Wear-Resistant and Conductive Probe with PtSi Tip

**DOI:** 10.3390/mi12111326

**Published:** 2021-10-28

**Authors:** Meijie Liu, Yinfang Zhu, Junyuan Zhao, Lihao Wang, Jinling Yang, Fuhua Yang

**Affiliations:** 1Institute of Semiconductors, Chinese Academy of Sciences, Beijing 100083, China; mjliu@semi.ac.cn (M.L.); junyuanzhao@semi.ac.cn (J.Z.); lhwang@semi.ac.cn (L.W.); fhyang@semi.ac.cn (F.Y.); 2College of Materials Science and Opto-Electronic Technology, University of Chinese Academy of Sciences, Beijing 100049, China; 3State Key Laboratory of Transducer Technology, Shanghai 200050, China

**Keywords:** ICP isotropic etching, annealing, wear-resistant, PtSi tip

## Abstract

This paper presents a simple and reliable routine for batch fabrication of wear-resistant and conductive probe with a PtSi tip. The fabrication process is based on inductively coupled plasma (ICP) etching, metal evaporation, and annealing. Si tips with curvature radii less than 10 nm were produced with good wafer-level uniformity using isotropic etching and thermal oxygen sharpening. The surface roughness of the etched tip post was reduced by optimized isotropic etching. The dependence of the platinum silicide morphology on annealing conditions were also systematically investigated, and conductive and wear-resistant probes with PtSi tips of curvature radii less than 30 nm were batch fabricated and applied for scanning piezoelectric samples.

## 1. Introduction

Scanning probe microscope (SPM) is one of the most powerful tools used in surface profile measurement, high density data storage, and bio-engineering [1,2,3,4,5,6,7,8,9]. Its core components are scanning probes [10,11,12]. Until now, various types of SPMs have been developed, among which electrostatic force microscope (EFM) [13,14] and piezoresponse force microscope (PFM) [15,16] are one of the most commonly used. In the EFM and PFM systems, conductive tips with high resolution, good electrical conductivity as well as high wear-resistance are highly desired [17,18].

Nowadays, various kinds of conductive probes have been studied including highly doped Si probes [19], carbon nanotube (CNT) probes [20,21,22], polymer graphite (PG) probes [23], and coated standard (Si or SiN) AFM probes [24,25,26,27,28,29,30,31,32]. Highly doped Si probes have relatively high conductivity and a sharp tip with curvature radii less than 10 nm. However, the tips are easily worn, and when they are exposed to air, the native oxide layer on the probe surface deteriorates its conductivity. The CNT probes are more durable than Si probes, and have comparable curvature radii and much higher aspect ratio. They are obtained either by attaching nanotubes to conductive AFM tips under optical or electron microscopes, or by growing nanotubes on conductive AFM tips directly [20,21]. However, nanotubes can only be attached onto individual tips. This method is not suitable for batch fabrication. Direct chemical vapor deposition (CVD) growth is currently the only way possible to produce CNT probes in batch, but the challenge is how to grow nanotubes anchored at the tip apex with designed orientation [22].The main characteristics of polymer graphite are high conductivity, considerable durability, and low price. Thus, a polymer graphite is suitable for tip production. The PG probes are usually fabricated by electrochemical etching combining with mechanical sharpening or focused ion beam milling [23]. This method is also not suitable for batch fabrication.

The coated probes, with metals such as Ti, Au or Pt [24,25,26], with alloys such as TiN or PtSi, with doped diamond [28,29,30,31,32], are the most common conductive probes. However, wear resistances of these metal-coated probes are poor despite their good conductivities. Doped diamond-coated probes have good wear-resistance because of excellent hardness, but poor conductivity compared to the metal-coated probes, and blunter tips with curvature radii above 100 nm [27]. Silicidation is a technique to improve the mechanical properties and wear resistance of noble metals [33,34]. PtSi probes with high electrical conductivity, relative high hardness [35], and small curvature radii are ideal conductive and wear-resistant SPM probes.

Different approaches have been reported to achieve PtSi probes based on MEMS manufacturing technologies. Depositing single PtSi nanoparticle on the bare Si tip apex of the AFM probe is suitable for fabricating a single conductive probe, but not suitable for batch fabrication [29]. Thermal treatment of Pt-coated Si probes is a commonly used method to form PtSi tips in batches. Large-scale manufacturing PtSi probes on silicon-on-insulator (SOI) wafers have been proposed, combining isotropic etching and oxidation sharpening to form a nanoscale Si tip, followed by selectively sputtering Pt around the tips and annealing to form the PtSi tip [30,31]. However, all these approaches have difficulties in achieving high uniform fabrication at the wafer-level.

In this work, the batch fabrication method of conductive probe with small curvature radii, good conductivity, and wear resistance is presented. The fabrication process for a nano-tip with nanometer curvature radii and high aspect ratio was optimized to achieve high uniformity at the wafer-level. The conductive probes were applied in the PFM and the scanning results for calibration grating (tipsnano, Tallinn, Estonia) samples were comparable to those of the commercial probes. The scanning results for lithium niobate (LiNbO_3_) single-crystalline samples (tipsnano, Tallinn, Estonia) show that the conductive probes were wear-resistant.

## 2. Design and Fabrication

Figure 1 describes the fabrication process of conductive probes. A 4-inch (100)-oriented SOI wafer was used. First, a 200 nm thick oxide was thermally grown on the substrate as the mask. The lithography was performed to define the tip apex, and the pattern was transformed to the thermal oxide by reactive ion etching (RIE), as illustrated in Figure 1a. Then, Si etching to a depth of 16 μm was followed by isotropic etching using ICP to obtain the Si post, as illustrated in Figure 1b. The cantilever was shaped by anisotropic etching using ICP. Thermal oxidation at 1000 °C was conducted to sharpen and protect the tip post (Figure 1c). A 3 μm thick oxide was deposited by plasma enhanced chemical vapor deposition (PECVD) on both sides of the wafer as the mask. Then, the Si substrate was etched to define the chip body from the backside by deep reactive ion etching (DRIE, SF_6_-based Bosch process) until a desired depth of about 350 μm, and an isotropy ICP was employed to remove the residual Si substrate from the front side of the wafer (Figure 1d). Subsequently, the cantilever was released by removing the oxide mask and buried layer using buffered hydrofluoric (BHF) (Figure 1e). Finally, a 30 nm Pt film was deposited on the front side of the wafer using electron beam evaporation and followed by annealing (Figure 1f). The scanning electron microscope (SEM) picture of the conductive probe are shown in Figure 8.

The tip morphology and curvature radii were examined by SEM. XRD was utilized to characterize the film phase components before and after annealing. Finally, the conductive probe was applied in PFM (CSPM5500, Guangzhou, China) to scan LiNbO_3_ single-crystalline samples to evaluate the conductivity and wear resistance.

## 3. Results and Discussion

Batch fabrication of Si nano-tips with good wafer level uniformity is technically challenging. Dependence of the etched tip profile on ICP isotropic etching parameters such as chamber pressure, gas flow rate, and lower electrode power were systematically investigated [36]. The ICP etching was optimized to reduce the roughness of the tip post. The recipe of pure SF_6_ isotropic etching is illustrated in Table 1, and the ICP power and SF_6_ gas flow rate were kept constant to exclude their effect on the etched surface roughness.

Figure 2 shows the tip profiles etched in different chamber pressures, where the tip surface roughness was reduced as the chamber pressure increased. In SF_6_-based ICP isotropic etching of Si, the density of fluorine radicals determines the etch rate and the shape of the tip post [36], and depends on ICP power, SF_6_ gas flow rate, and chamber pressure. The chamber pressure not only affects the density of fluorine radicals, but also affects its mean free path [37]. The density of fluorine radicals increases with chamber pressure while the SF_6_ gas flow rate remains constant, which results in a higher etching rate. Moreover, the higher density leads to more collisions between fluorine radicals, thus smaller mean free path and larger probability of the lateral movement of fluorine radicals. As a result, enhanced fluorine radical bombardment on the sidewall of the tip post removes the micro-mask from the tip surface and allows the fluorine radicals to react sufficiently with the Si atoms on the tip side. Thus, increasing the chamber pressure effectively reduces the tip surface roughness.

When the chamber pressure is set at 1.6 Pascal, the SF_6_ gas flow rate is 100 sccm, the ICP power is 1000 W, and the tip post with a smooth surface is obtained at the wafer-level.

In ICP isotropic dry etching, large exposure ratio leads to severe edge effects. The etching rate is faster at the edge than in the center, which cannot be fully compensated by mask compensation [36]. After mask compensation, the remaining neck width of the tip post after etching was still in the range of 490 nm to 555 nm, as shown in Figure 3. Further thermal oxygen sharpening is required to obtain nano-tips with a curvature radii less than 10 nm.

Thermal oxidation is employed for tip sharpening. Oxygen diffuses through the SiO_2_ layer and reacts with Si atoms to form new oxide layers. The new oxide layers are constantly being generated at the SiO_2_–Si interface. The volume of the Si atom is 20 Å^3^ and the volume of the SiO_2_ molecule is 45 Å^3^ [38]. The mismatch of the volume gives SiO_2_ compressive stress, which decreases the diffusion constant of oxygen. The stress at the Si/SiO_2_ interface around the convex or concave corner is difficult to release, thus slows down the oxidation rate and leads to tip sharpening [39]. The glass transition temperature of SiO_2_ is 960 °C [40]. When the oxidation temperature is below 960 °C, SiO_2_ viscous flow is weak. The stress at the SiO_2_–Si interface at the neck of the tip post is not easily released, which makes it impossible to completely remove defects, and results in distorting or fracturing the Si nucleus inside the tip neck under stress. When the oxidation temperature is above 1050 °C, the SiO_2_ viscous flow is enhanced. The stress at the Si/SiO_2_ interface can be easily released. The tip surface defects can be removed based on the Si corner rounding effect (CRE) [41] and the enhanced oxide flow. However, excessive oxidation rate at higher oxidation temperature leads to larger curvature radii and smaller tip heights.

In order to make a tip with a smooth surface and small curvature radii, thermal oxidation sharpening was carried out at 1000 °C with a thermal oxide thickness of 540 nm. As shown in Figure 4, the tip after oxidation sharpening has the curvature radii of 5 nm. The size distribution of the curvature radii along the wafer after oxidation is shown in Figure 5. When the neck width of the tip post was about 520 nm, the tip had the lowest radii curvature after oxidation sharpening. The tip post with the 520 nm neck width was only sharpened, as the oxide thickness in the plane was 540 nm. Over-oxidation occurs when the size of the tip neck was less than 520 nm, and under-oxidation occurs when the size of the tip neck is more than 520 nm. Thus, the curvature radii of the tips were a little bit larger.

The tip apex had curvature radii less than 10 nm, indicating that thermal oxidation sharpening can clearly improve the tip apex uniformity and fabrication yield up to 90%.

PtSi films are conventionally produced using co-evaporation [42], co-sputtering [43], or solid-phase reaction by annealing a Pt thin film deposited on the Si substrate [17,44,45]. The solid-phase reaction is more appropriate for the preparation of conductive probes, can ensure better adhesion to the tip surface, and a thicker conductive layer with a sufficiently small curvature radii of the tip [17]. Pt atoms can be supplied by diffusion from Pt film to the growing silicide layer and form silicide during annealing [44].

There are many platinum silicides, among which Pt_2_Si and PtSi are stable phases. As the activation energy of the Pt_2_Si phase is 28.8 kJ/mol and that of PtSi is 33 kJ/mol, the mixed atoms at the interface between Si and Pt easily form metastable Pt_2_Si, which are then transformed into PtSi [45]. Below 300 °C, the Pt_2_Si phase forms, while the PtSi phase grows above 300 ℃. As given in Table 2 for different annealing recipes, in this work, the annealing temperature rises in the sequence of 200–300–500/550 °C to fabricate PtSi probes.

Figure 6 shows the XRD spectra of Pt/Si films before and after annealing in different conditions. The Pt/Si films annealed with recipe 1 formed PtSi and Pt phases, no Pt_2_Si phase was observed. This indicates that the Pt_2_Si phase was completely converted to PtSi, but the remaining Pt hardly reacted with Si to form Pt_2_Si.

The Pt_2_Si thickness can be estimated by [45]:$$X^{2}=Dt$$
where *X* is the film thickness; *D* is the diffusivity at annealing temperature; and *t* is the annealing time. As the annealing furnace is not perfectly sealed, impurities, especially oxygen, are inevitably present in a N_2_ atmosphere. During Pt_2_Si formation, oxygen atoms diffuse to the Pt layer and react with the Si atoms, forming an oxide layer above the Pt_2_Si layer. The oxide layer hinders Pt diffusion [46], thus the growth rate of Pt_2_Si is reduced, while the formed Pt_2_Si is unaffected by the oxide layer. The rate of Pt_2_Si consumption is larger than that of Pt_2_Si formation. As a result, complete conversion of Pt_2_Si to the PtSi phase takes place, and there was only the PtSi phase and the unreacted Pt phase in the annealed film.

When annealing the Pt/Si films in a forming gas (N_2_:H_2_ = 5:1), the reducing hydrogen atmosphere can minimize premature oxidation and oxygen diffusion into the Pt film, which can promote the formation of Pt silicide [47]. Therefore, in an annealing atmosphere referring to recipes 2 to 4, Pt completely reacted with Si and was converted to PtSi.

The tips covered with Pt films are annealed in the conditions referring to recipes 2 to 4, and their surface morphologies are shown in Figure 7. The surface morphology of the PtSi films on the tip was flat and continuous when the annealing temperature was low and annealing time was relatively short (recipe 4). The Pt atoms adjacent to the interface of Pt/Si reacted with Si atoms and Pt_2_Si was formed first. As Pt atoms continue to diffuse, Pt reacts with Pt_2_Si to form PtSi. The process causes a volume change. The strains due to the volume change cannot be relaxed, which leads to the vertical growth rate of PtSi grains being greater than the lateral growth rate. Thus, PtSi prefers to form columnar grains [48]. When the annealing temperature increases, the PtSi columnar grains grow and aggregate, resulting in a raised roughness on the film surface, and even fracture [47].

When the PtSi film fractures, the conductivity of the tip deteriorates. In order to ensure that the PtSi tip has both good wear resistance and good electrical conductivity, the deposited Pt film probe was annealed in optimum conditions (recipe 4). The resulting SEM image of the wear-resistant and conductive probe is shown in Figure 8. The curvature radii of PtSi tip was 27 nm.

The probe was applied in a PFM to scan the calibration grating (tipsnano, Tallinn, Estonia). As shown in Figure 9, the fabricated conductive probes achieved good imaging resolution comparable to that of the commercial probes. The probe was also used to scan the LiNbO_3_ single-crystalline samples with a scanning area of 5.5 μm × 5.5 μm. The 1000th scanning image was compared with the 1st scanning image, as shown in Figure 10. After 1000 scans, the fabricated conductive probe still had good conductivity, indicating that the fabricated conductive probe with the PtSi probe had good wear resistance.

## 4. Conclusions

This work presents a batch fabrication method for a wear-resistant and conductive probe with a PtSi tip. The fabrication process is based on inductively coupled plasma (ICP) etching, metal evaporation, and annealing. Si tips with a curvature radii less than 10 nm at the wafer-level was fabricated with good uniformity by optimized isotropic ICP etching in combination with thermal oxidation sharpening. Step annealing at 200 °C for 10 min, 300 °C for 10 min, and 500 °C for 45 min in forming gas achieved the best PtSi tip morphology. PtSi tips with good conductivity, wear resistance, and a curvature radii less than 30 nm was produced, which has imaging resolution comparable to that of the corresponding commercial probes.

## Figures and Tables

**Figure 1 micromachines-12-01326-f001:**
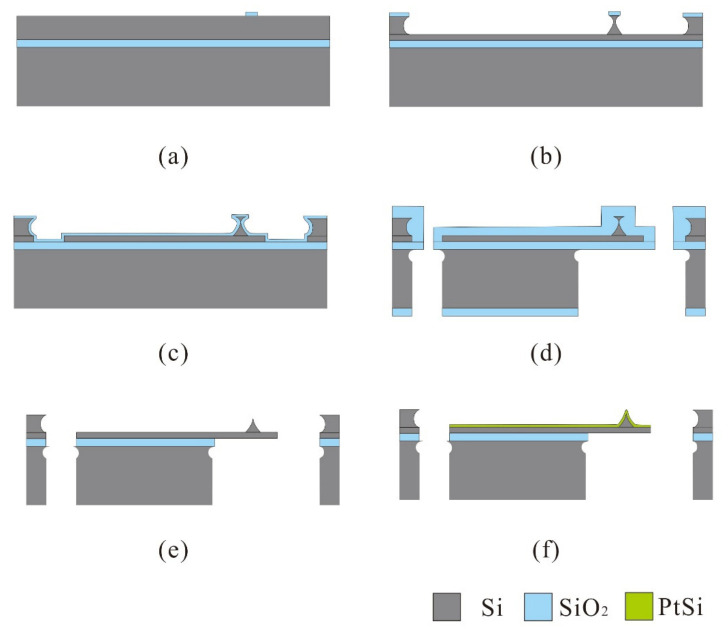
Fabrication process of conductive probes. (**a**) Pattern oxide layer as the mask. (**b**) Isotropic etching of silicon. (**c**) Thermal oxygen sharpening. (**d**) PECVD SiO_2_ as mask and define the chip body. (**e**) Removal the oxide layer in BHF. (**f**) Depositing Pt films and annealing.

**Figure 2 micromachines-12-01326-f002:**
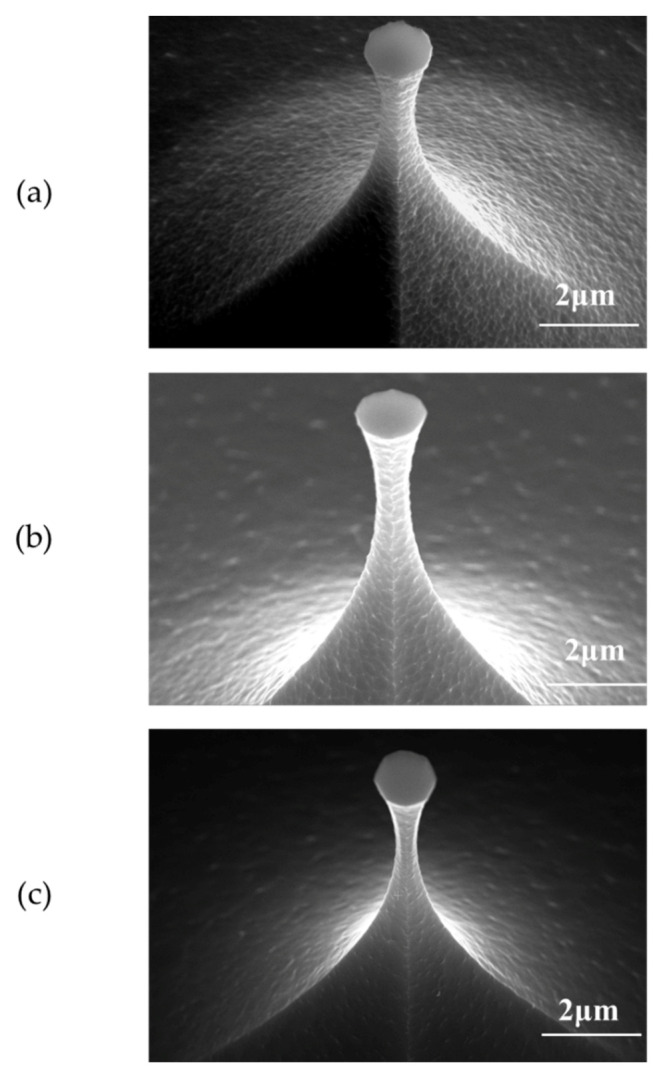
The profiles of the Si tip post etched in different chamber pressures: (**a**) 1.1 Pascal, (**b**) 1.3 Pascal, (**c**) 1.6 Pascal.

**Figure 3 micromachines-12-01326-f003:**
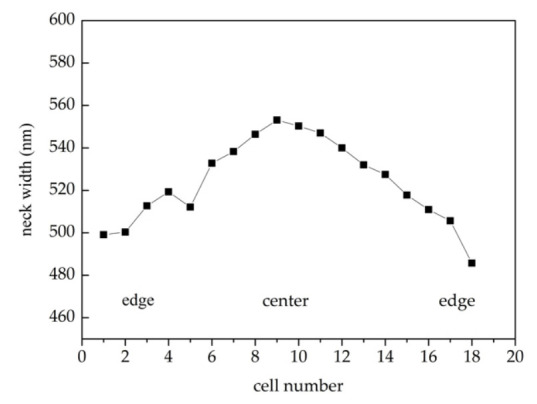
The neck width of the tip post at different positions.

**Figure 4 micromachines-12-01326-f004:**
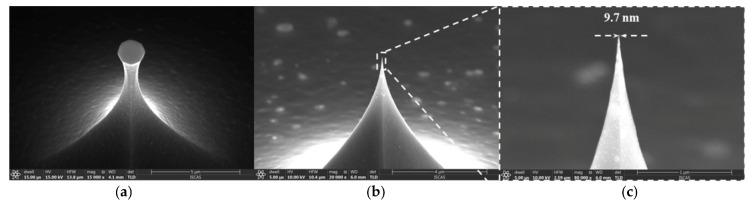
Si nano-tip profile: after etching (**a**) and after oxidation sharpening (**b**,**c**).

**Figure 5 micromachines-12-01326-f005:**
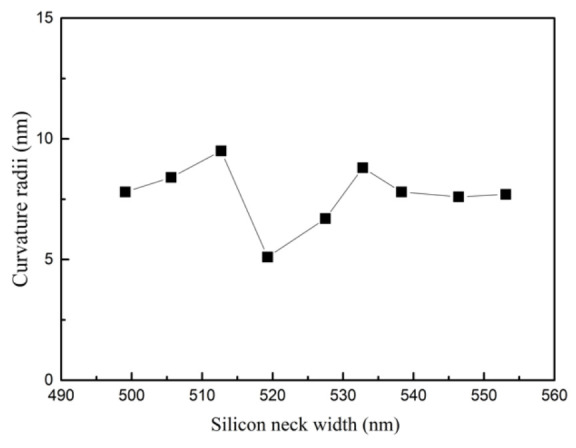
Curvature of the tip apex for the tip post with different neck widths.

**Figure 6 micromachines-12-01326-f006:**
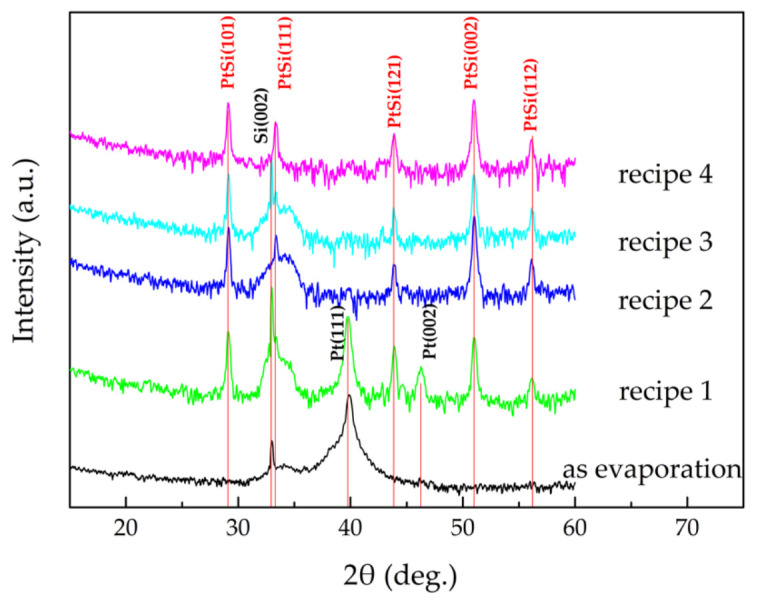
XRD spectra of the Pt/Si films before and after annealing in different conditions referring to the recipes in Table 1.

**Figure 7 micromachines-12-01326-f007:**
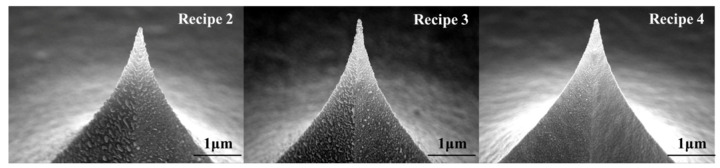
Tip morphology after different annealing conditions.

**Figure 8 micromachines-12-01326-f008:**
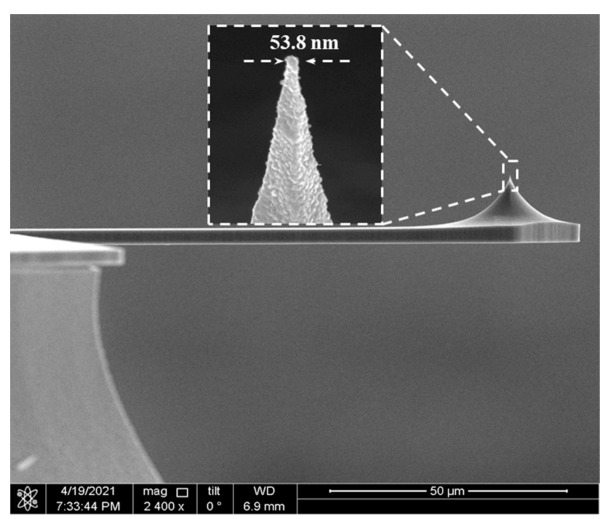
SEM image of the conductive probe.

**Figure 9 micromachines-12-01326-f009:**
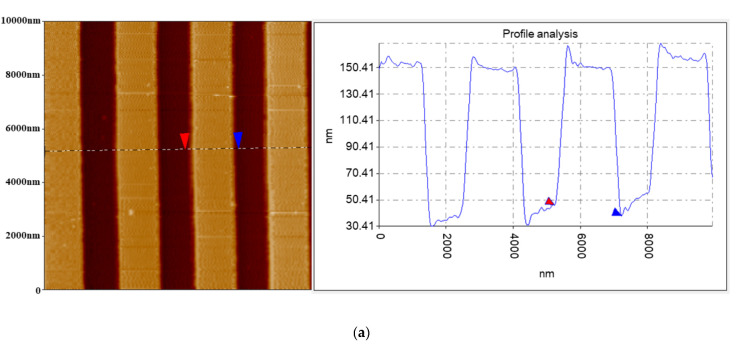
The scanning results for calibration grating (tipsnano, Tallinn, Estonia) (**a**) using the fabricated probe and (**b**) using a commercial probe.

**Figure 10 micromachines-12-01326-f010:**
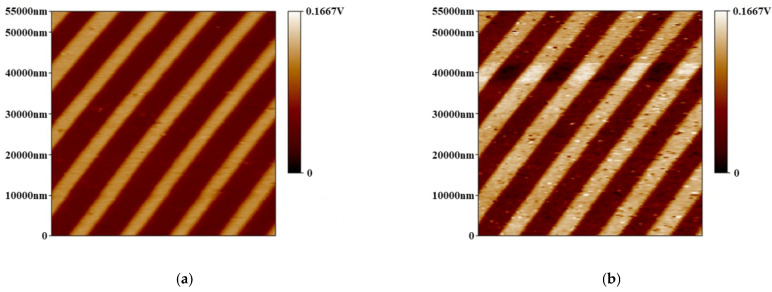
Comparison of the scanning results for the LiNbO_3_ single-crystalline samples: (**a**) the first time and (**b**) the 1000th time.

**Table 1 micromachines-12-01326-t001:** ICP etching parameters.

Parameter	ICP Power (W/MHz)	SF_6_ Flow Rate (sccm)	Pressure (Pascal)	Etching Time (sec)	Neck Width (nm)
I	1000/2	100	1.1	270	863
II	1000/2	100	1.3	267	701
III	1000/2	100	1.6	268	478

**Table 2 micromachines-12-01326-t002:** Annealing recipes.

Annealing Recipe	Annealing Atmosphere	Temperature/Time
1	N_2_	200–300–550 °C 10 min–10 min–90 min
2	N_2_:H_2_ = 5:1	200–300–550 °C 10 min–10 min–90 min
3	N_2_:H_2_ = 5:1	200–300–550 °C 10 min–10 min–60 min
4	N_2_:H_2_ = 5:1	200–300–500 °C 10 min–10 min–45 min

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
