# Peer review of "Batch Fabrication of Wear-Resistant and Conductive Probe with PtSi Tip"

_micromachines, 2021, doi:10.3390/mi12111326_

Round 1
Reviewer 1 Report
Dear authors,
thank you very much for the nice study. The paper is well written and results are really encouraging. In general, you could elaborate the experimental design a bit more than in the current state in order to make it easier to verify by other studies. Apart from that, I have put some comments directly in the pdf file, which could help to further improve the quality of the manuscript.
All the best.
Reviewer

Reviewer 2 Report
Batch fabrication of wear-resistant and conductive probes for scanning probe microscopy (SPM) is a long-standing problem limiting many aspects in the development and applications of the technique. The authors of this manuscript present an approach that seems viable and can be of significant interest to the SPM community. The paper is well-written, and I recommend its publication. My only remark is that the manuscript needs careful proofreading. There are multiple minor spelling and grammar problems in the text.
Reviewer 3 Report
As for the introduction part (1), the indtroduction provide rather sufficient background but it is reccomended that authors should mention the Polymer graphite pencil lead used as a cheap alternative for classic conductive SPM probes, which was published recently in Nanomaterials journal recently.
The design and fabrication part (2) should provide more precise information about the fabrication process so it could be reproduced by other researchers. Please provide more detailed information for each technology step made. The section is very short.
The Results and Discussion part (3) provide only few places where corrections are needed: the pressure should be stated in Pascals or Milibars instead of Torrs (fig 2 also); as for the figure 2, the information bar is not adequatly readable in this size so I would suggest cropping in and provide only the scale directly to the image; as for the ICP isotropic dry etching, the edge effects should be described at least in rough shape; the new oxide layers mentioned on line 114 should be described in more details in order to judge the parameters and qualities/setbacks of the SiO2-Si interface.
In line 176 authors write: The Pt atoms adjacent to the interface of Pt/Si first react and form silicides, and then go through the layer.. can you be more specific about the movement through the layer? What is preciselly happening there.
On page 7/10, the quality of the figure 8 is not sufficient. The axes cannot be read properly since the numbers and letters are very small. The same for the profile analysis parts located bottom, please correct this along with figure 9., where there is same problem.
